# Addressing Healthcare Gaps in Sweden during the COVID-19 Outbreak: On Community Outreach and Empowering Ethnic Minority Groups in a Digitalized Context

**DOI:** 10.3390/healthcare8040445

**Published:** 2020-11-01

**Authors:** Giuseppe Valeriani, Iris Sarajlic Vukovic, Tomas Lindegaard, Roberto Felizia, Richard Mollica, Gerhard Andersson

**Affiliations:** 1Refugee Health Centre, Ostergotland County Council, 601 82 Norrköping, Sweden; roberto.felizia@regionostergotland.se; 2Department for Affective Disorders, Sahlgrenska University Hospital, 413 45 Gothenburg, Sweden; iris.sarajlic.vukovic@vgregion.se; 3Department of Behavioural Sciences and Learning, Linköping University, 581 83 Linköping, Sweden; tomas.lindegaard@liu.se (T.L.); gerhard.andersson@liu.se (G.A.); 4Harvard Program in Refugee Trauma, Massachusetts General Hospital, Harvard Medical School, Cambridge, MA 02115, USA; richard.mollica@gmail.com

**Keywords:** COVID-19, health gaps, e-Health, social determinants of health, empowerment, public health

## Abstract

Since its early stages, the COVID-19 pandemic has interacted with existing divides by ethnicity and socioeconomic statuses, exacerbating further inequalities in high-income countries. The Swedish public health strategy, built on mutual trust between the government and the society and giving the responsibility to the individual, has been criticized for not applying a dedicated and more diverse strategy for most disadvantaged migrants in dealing with the pandemic. In order to mitigate the unequal burden on the marginalized members of society, increasing efforts have been addressed to digital health technologies. Despite the strong potential of providing collective public health benefits, especially in a highly digitalized context as Sweden, need for a stronger cooperation between the public health authorities and migrant community leaders, representatives of migrant associations, religious leaders and other influencers of disadvantaged groups has emerged. Suggestions are presented on more culturally congruent, patient-centered health care services aimed to empower people to participate in a more effective public health response to the COVID-19 crisis.

## 1. Introduction

The emergence of novel coronavirus SARS-CoV-2 infection and its related disease (COVID-19) has caused a global public health emergency. As of October 2020, over 40 million infections have been reported worldwide, and COVID-19-related deaths passed 1 million worldwide [1]. The effect of COVID-19 has extended beyond health care, having significant social, economic, and cultural ramifications. Since its early stages, international findings have suggested that men [2], the elderly [3], racial and ethnic minorities [4], and people occupying lower socioeconomic positions [5] are more prone to developing severe COVID-19, or dying from it. In consideration of the widely adopted message that COVID-19 “does not discriminate”, one given belief by the Director-General of the World Health Organization (WHO) [6], these patterns have raised public health concerns. Socio-demographic analyses from several national Public Health Authorities provide increasing evidence on how the interaction of the virus and its environment does discriminate, exerting an unequal burden on the most disadvantaged members of society. In Sweden, the national Public Health Agency (Folkhälsomyndigheten) has expressed particular concern regarding the impact of COVID-19 on ethnic minorities [7]. 

The primary goal of our work is to show the weight of the relation between foreign country of birth, socio-economic status and the pandemic’s impact in Sweden through access to data from the Swedish authorities. Secondly, we present ongoing strategies to mitigate the consequences of COVID-19 on migrants and, ultimately, we discuss further challenges in order to bridge the gap between public health and health care delivery to marginalized populations.

## 2. The Swedish Context and the Impact of the Pandemic: Socio-Demographic Data

Sweden is often portrayed as one of the most prominent representatives of an officially declared multicultural policy [8]. Indeed, Sweden has been celebrated, alongside Canada and Australia, as a model for positive multicultural immigrant integration, by considering cultural diversity a public good and a potential source of richness for civil society [9,10]. Today, 19.6% of the population is foreign-born, with a rapid growth in recent years as a result of several waves of refugees from various conflict areas of the world [11]. Recent years have seen an increase in immigrants in Sweden, and only in 2015, more than 160,000 people applied for asylum in Sweden, corresponding to 1.6% of Sweden’s population [12]. The major groups of asylum seekers applying for permits to stay in Sweden since 2012 were refugees from Syria, Afghanistan, Iraq, Somalia, Eritrea and those who were stateless (e.g., coming from Palestine) [13].

Over the last decades, Sweden has struggled with issues concerning housing segregation, discrimination and integration of immigrants and refugees into the labor market and in society at large. In April 2018, the Swedish Parliament stated that they wished to eliminate “health gaps among ethnic groups within a generation through an active and strengthened public health policy” [14]. Contrary to this perspective, the COVID-19 outbreak further emphasized the existing healthcare gaps between Swedish-born and foreign-born people, a danger that has been deeply debated in the media as well as in the scientific literature, along with the whole Swedish public health strategy during the COVID-19 crisis. 

Traditionally, Sweden has been referred to as a society of state individualism, i.e., an eclectic mix of deeply individualistic behaviors and a strong state and governance [15]. In accordance with these principles, even in the pandemic, Sweden opted to rely on an approach of explicit cooperation between the state’s response and people’s individual responsibility to reduce the spread [16]. At the international level, although Sweden realized the same preventive interventions as most other countries accordingly to WHO recommendations, working to flatten the epidemiological curve by slowing transmission so that the healthcare system could cope with the disease, the absence of compulsory lockdown measures led to misinterpretations and narratives of a controversial “Swedish experiment” [17]. 

Overall, Sweden has had a similarly intermediate number of reported COVID-19 deaths—fewer per capita than Spain, the United Kingdom and Italy that adopted late-onset stringent mandates, but significantly more than its Scandinavian neighbors who implemented strong measures promptly and more than most other European countries. As of October 2020, the Swedish National Board of Health and Welfare has registered over 100,000 infection cases and 5930 deaths [18]. The overlay of mutual trust with the authorities and individual responsibility about preventive measures could substantially avoid an overwhelming of the healthcare system; still, increasing concerns have been raised about the effectiveness of Swedish strategy on disadvantaged ethnic groups. Table 1 shows data reported by the Swedish Public Health Authority on the relation between COVID-19 cases and country of birth over the period 13 March–7 May, when the total number of cases was 24,623 and the total number of deaths related to COVID-19 was 3040 [7]. 

The incidence of COVID-19 cases was highest among those born in Turkey with 753 per 100,000 people, followed by Ethiopia with 742 per 100,000 people and Somalia with 660 per 100,000 people. The incidence among those born in Sweden was 189 per 100,000 people. With respect to the female gender in relation to the country of birth, the incidence was highest among those born in Ethiopia with 792 per 100,000 people followed by Somalia with 725 per 100,000 people. For men, the incidence was highest among those born in Turkey with 820 per 100,000 people followed by Ethiopia with 693 per 100,000 people. Among the Swedish-born population, a relative higher risk was found in women versus men, 226 and 152 per 100,000 people, respectively. 

With respect to median age of deceased persons, a significant difference was found between the group born in Somalia and the Swedish-born people, with median ages of the deceased of 68 and 85 years old, respectively (see Table 2). Starting from these data, Drephal et al. [19] examined multivariate variation in the risk of death from characteristics that include age, sex, education level, disposable income, and country of birth. Immigrants from low- and middle-income countries displayed 2.5 times higher mortality among men (HR_Men_: 2.56; 95% CI: 2.18, 3.01) and more than 1.5 times higher among women (HR_Women_: 1.66; 95% CI: 1.32, 2.09) as compared to those born in Sweden. On the other hand, immigrants from high-income countries only displayed 20% higher mortality among men (HR_Men_: 1.19; 95% CI: 1.01, 1.39) and nearly 10% higher among women (HR_Women_: 1.08; 95% CI: 0.92, 1.26) compared to the Swedish-born group. In spite of this, the highest incidence of deaths related to COVID-19, 145 cases per 100,000 persons, was registered in the Finnish-born group: a possible explanation is the high proportion of Finnish-born people aged 70 years old and above facing the issue of virus spread in retirement homes. With respect to education and net of income, a gradient for both men and women was found with individuals, with primary education experiencing approximately 25% (HR_Men_: 1.23; 95% CI: 1.07, 1.42) and 50% (HR_Women_: 1.48; 95% CI: 1.25, 1.75) higher mortality, respectively, than individuals with post-secondary education.

The same researchers [19] found substantial differences in the demographic risk factors between working age people (ages 65 and below) and retirees (ages 66 and higher). In working ages, compared to retirees, male gender, education and income were stronger predictors of dying from COVID-19. Particularly alarming was the finding that subjects in the lowest income tertile had more than five times higher risk of dying from COVID-19 (HR: 5.21; 95% CI: 3.38, 8.02) compared to those in the highest, and those with primary (HR: 2.63; 95% CI: 1.66, 4.18) and secondary (HR: 2.02; 95% CI: 1.27, 3.22) education had more than two times higher risk compared to those with post-secondary education. Socio-economic differentials in COVID-19 mortality were much less pronounced among individuals in retirement ages.

In another statistical analysis by Hansson et al. [20], considering all mortality cases in Sweden in the same period (March to May 2020), an excess mortality emerged in people born in low- and middle-income countries compared with people born in Sweden, Scandinavia and North-America. Compared to the same months over the period of 2016–2020, an excess mortality of 220% was found among people born in Somalia, Iraq and Syria both in the age range of 40–65 and aged 65 years or older. Conversely, among people born in high-income countries (Sweden, Scandinavia and North America), a slight decreased mortality, 1%, was found among persons aged 40–65 years, and an excess mortality of 19% of those aged 65 and above. 

Based on these findings, the Swedish strategy has been criticized for not applying a dedicated and more diverse strategy for disadvantaged migrants in dealing with the COVID-19 pandemic. 

Immigrants from Africa and Middle East living in Sweden face difficulties of conducting voluntary social distancing in settings with household overcrowding, they depend more on public transport, and cover a large proportion of service sector jobs, often lower paying, such as public transportation operators or grocery store or pharmacy clerks. This makes them more exposed to the public and thus to being infected. Furthermore, they face language and cultural barriers, limiting their access to accurate information on prevention and mitigation, compelling them to rely on social media to obtain advice that may be erroneous. People with poor access to healthcare who experience COVID-19 related symptoms may delay or even forgo being tested, and may consequently turn to medical care only in advanced stages, resulting in poorer outcomes [21]. This may potentially put their families and communities at risk as well. 

## 3. Digital Health: Bridging Distances and Empowering People

The COVID-19 outbreak poses immense challenges, demanding reduction of healthcare inequities at a rapid pace. Preparedness and response plans at national and subnational levels need to follow WHO’s guiding principles: speed, because of the explosive nature of the virus; scale, because everyone in society has a part to play in building the capacities required to control this pandemic; and equity, because an inclusive approach that leaves no one behind should guide our public health efforts [22]. To coordinate mechanisms for the urgent implementation of public health capacities, digital health technologies have been identified as one of the most promising approaches. It has been proposed that using digital health strategies can aim at resolving “the potential crisis in the provision of health services to helping preserve and reconstruct a post-pandemic society” [23]. 

Sweden is a highly connected society and one of world’s leading countries in the development of the net economy as well as mobile communications. In 2016, 93% of inhabitants had Internet access, 79% made online purchases and 77% had smartphones or portable computers—all well above the average for the European Union [24]. In the same year, Sweden’s government and the Swedish association of local authorities and regions endorsed a shared vision of making the country a global leader in e-health by 2025. More than 90% of pharmaceutical prescriptions are electronically transmitted (e-prescriptions), generated in the doctors’ electronic prescription system and transmitted through a secure network to the national e-prescription database. The market for virtual doctors is also expanding, with a rapid growth in the number of digital health companies and initiatives using voice, video and text services to cut health-care providers’ costs. 

Although the development of the vision e-Health 2025 encountered a number of bottlenecks during the past years [25], e.g., security and privacy issues and need to set up clear agreements among key parties in the health care sectors, the COVID-19 outbreak has meant an unprecedented upswing for digital care in several Swedish counties. On 12 March 2020, the Healthcare Guide online, 1177.se, which is Sweden’s national hub for advice, information, inspiration and e-services for health and healthcare, registered the record high of 1.6 million patient calls, compared to 200,000 an ordinary day. Only in Stockholm County Council, with about 2.3 million inhabitants, 1177.se registered about 7 million contacts in March 2020. In January and February 2020, the same service had less than five million contacts per month, and about four million in the same period last year. Yet, in Stockholm, the number of medical visits via video-links was 36,000 during April 2020, while in January 2020 only 3000 [26]. In that period, digital health tools played a pivotal role in facilitating the delivery of essential health care in Sweden and in empowering people to play an important and active role in limiting the spread of COVID-19. 

Despite the promising effectiveness of the digital health approach in a large part of the population, the COVID-19 crisis clearly exemplifies the impossibility of establishing a single technological solution to a given complex problem. While a number of crucial challenges, related to pre-existing inequalities such as income, housing and neighborhood density, require response plans by the highest levels of policy-making, other urgent issues regard digital health implementation. As observed by Fagherazzi et al. [27] acceptability of digital solutions may face challenges due to potential conflicts with users’ cultural backgrounds, calling for further efforts to improve community outreach and engagement of ethnic minority groups. Consistent with this, since the end of April 2020, several healthcare centers in Stockholm, Gothenburg, Norrkoping and other main cities in Sweden implemented telemedicine platforms in a culturally and linguistically appropriate direction, through so called “corona lines” [28]. They consist of phone line services that enhance the existing national phone lines to reach a considerable number of migrants. Healthcare professionals fluent in different languages (Arabic, Somali, Tigrinya/Amharic and Persian/Dari) are available to provide information about preventive measures, triage of those with respiratory symptoms as well as about home care and individual and community hygiene. The possibility to share native language and a similar cultural background with immigrants and refugees gives the professionals working in these services the opportunity to better respond to the health needs of these populations in an accurate, timely and user-friendly way [29]. Investments in such initiatives addresses four main goals: (1) to raise awareness among ethnic minority groups regarding the needed changes in habits due to the pandemic; (2) to decrease migrants’ barriers to access to health services (e.g., uncertainty regarding legal entitlements to healthcare, the fear of being reported to police and deported, and language difficulties); (3) to give support that strengthens social cohesion, solidarity, and healthy coping, and reduces loneliness; and (4) to empower patients to participate in the health care decision-making process and by increasing trust and reducing skepticism and the sense of alienation with the local authorities. Parallel lines of interventions as efforts in targeting specific communities have been established (e.g., multilingual pamphlets, videos shared on Facebook by community groups, posters in community centers etc.), and, thus, the corona lines could spread and enhance the provision of information among minority populations. In health care providers’ experience, liaising with local religious leaders and other influencing members in the communities of the minority groups has been essential in order to facilitate access to these services to most marginalized migrants [30]. 

There are still no available data from the Swedish Public Health Agency, after 7 May 2020, on the impact of the pandemic among foreign-born people. In our opinion, these initiatives promise not to reverse but to mitigate concerns of health inequities during these challenging times. New data are expected and will be available on the Swedish Public Health Agency’s official website in the near future.

## 4. Considerations and Future Directions

Contrary to the often-used phrase that “a virus does not discriminate”, the early stages of COVID-19 outbreak have pointed-out pre-existing inequalities among marginalized communities around the world [31]. The presented data, collected by the Swedish Public Health Authority, confirm similar trends to those reported in the scientific literature across the USA and other countries. WHO as well as Migrant and Ethnic Health Section of the European Public Health Association and other international organizations encouraged all health professionals to act together in order to improve community engagement of migrants in the campaign to face the health threats posed by COVID-19 [32]. At the subnational level, it means to meet the urgent challenges on how to improve communication and counter xenophobia; to provide culturally and linguistically appropriate, accurate, timely and user-friendly information in accessible formats on health facilities; and train groups of professionals and paraprofessionals to be able to communicate well with refugees and migrants. 

COVID-19 can be considered as the first true global epidemic of this magnitude in the digital era. Digital health solutions ought to be considered as valuable part of an effective pandemic response [33]. Telemedicine platforms are ideal for managing several challenges, for example, addressing the needs of low-acuity patients with disease exposure concerns, thus mitigating and preventing overcrowding in emergency departments, urgent care clinics, and primary care clinics. Telemedicine can also be used to address the ongoing healthcare needs of patients with chronic illnesses to reduce in-person clinic visits [34]. Such uses of telemedicine reduce human exposures (among healthcare workers and patients) to a range of infections. Digital tools can effectively support institutions during a pandemic by tracking virus transmission in real time [35]. Moreover, digital health strategies can facilitate the immediate widespread distribution of information, promote community engagement and contribute to the empowerment of people in adopting preventive measures and healthy life style decisions [36]. 

Despite the strong potential of providing collective public health benefits, especially in a highly digitalized country as Sweden, several challenges of mitigating the increasing health inequalities have emerged. Without underestimating other crucial social determinants of health among migrants, such as household crowding, poor health conditions and low socio-economic status, one lesson to take away during the pandemic in Sweden is to enhance cooperation between the public health authorities and migrant community leaders, representatives of migrant associations, religious leaders and other influencers of disadvantaged groups. Many suggestions have been put forward on how to focus on patient-centered care and a more culturally congruent health care environment. Culturally congruent, patient-centered care aims to improve health outcomes by empowering people to participate in the health care decision-making process and encompasses the following: respect for patients; understanding of culture, economic and educational status; and collaborative communication strategies [37]. The use of trained health professional sharing the cultural background of minority groups may help address some of these issues. Another commonly suggested strategy is the cultural adaptation of health care services, which refers to the systematic modification of an intervention to make it more aligned with cultural beliefs, values, language, context and/or behaviors of a specific cultural group [38]. This can also include factors that contribute to engagement and adherence, for example, targeted outreach efforts in community settings or offering treatment in more accessible and less stigmatizing ways [39].

It is becoming increasingly clear that the COVID-19 crisis is rapidly reinforcing health disparities drawn by historical and contemporary inequities. In addition, this crisis may leave legacies that will affect inequalities in the long term due to severe reduction in economic activity of many sectors. Without a well-thought-out policy response, respecting principles of solidarity, human rights, and equity, the post-COVID world could see inequalities worsening further [40]. Digital approaches have demonstrated, e.g., in South Korea [41], usefulness and reactivity in facing the immense challenges throughout the COVID-19 crisis. However, this challenging time demands further urgent needs to make meaningful practice implementations to address disparities in health and healthcare. At this time, by accessing to socio-demographic analyses limited to 7 May 2020, we are unable to assess on the effectiveness of previously described culturally adapted initiatives that implemented the Swedish public health strategy during the first wave of COVID-19. Nonetheless, we believe that striving towards patient centeredness and cultural adaption of health services and digital health tools, tailored to accommodate cultural differences in health-related values and beliefs, are key elements of delivering quality care to migrants. As Einstein stated, ‘‘in the midst of every crisis, lies great opportunity’’: by investing in such interventions now, we may mitigate disparities related to the current crisis and be better positioned to ensure more equitable health care in the future.

## Figures and Tables

**Table 1 healthcare-08-00445-t001:** Report by Swedish Public Health Agency (Folkhälsomyndigheten) on the number of cases and incidence of COVID-19 per country of birth during the period of 13 March–7 May 2020 [7].

Country of Birth	No. Cases	No. in the Population	% Cases	% in the Population	Incidence (Per 100,000 People)
Afghanistan	214	58,780	0.90	0.60	364
Bosnia and Hercegovina	206	60,012	0.90	0.60	343
Chile	175	28,025	0.80	0.30	624
Eritrea	218	45,734	0.90	0.40	477
Ethiopia	161	21,686	0.70	0.20	742
Finland	744	144,561	3.20	1.40	600
Germany	137	51,436	0.60	0.50	266
Iran	418	80,136	1.80	0.80	522
Iraq	876	146,048	3.80	1.40	600
Former Yugoslavia	287	64,349	1.20	0.60	446
Lebanon	152	28,508	0.70	0.30	533
Norway	105	41,578	0.40	0.40	253
Poland	170	93,722	0.70	0.90	181
Somalia	463	70,173	2.00	0.70	660
Sweden	15,676	8,307,856	67.90	80.40	189
Syria	594	191,530	2.60	1.80	310
Thailand	106	43,556	0.50	0.40	243
Turkey	389	51,689	1.70	0.50	753

**Table 2 healthcare-08-00445-t002:** Report by Swedish Public Health Agency (Folkhälsomyndigheten) on the number of deceased by country of birth, median age and incidence (number of deaths per 100,000 people), during the period of 13 March–7 May 2020. Countries with fewer than 11 deaths are not reported [7].

Country of Birth	No. Deceased	Median Age Deceased	Incidence (Per 100,000 People)
Bosnia and Hercegovina	21	82	35
Chile	18	79	64
Finland	210	82	145
Germany	32	88	62
Iran	45	83	56
Iraq	65	79	45
Former Yugoslavia	36	78	56
Lebanon	18	75	63
Norway	23	88	55
Poland	17	88	18
Somalia	52	68	74
Sweden	2678	85	32
Syria	67	76	35
Turkey	50	81	97

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
