# Peer review of "Addressing Healthcare Gaps in Sweden during the COVID-19 Outbreak: On Community Outreach and Empowering Ethnic Minority Groups in a Digitalized Context"

_healthcare, 2020, doi:10.3390/healthcare8040445_

Round 1

Reviewer 1 Report

I would like to congratulate the authors the current manuscript, but some minors changes are required:

  1. The authors should included the scientific named gives to Covid-19: SARS-CoV-2.
  2. The key points or definition of a positive multicultural immigrant integration  should be included.
  3. Examples should be included in some sentence, i.e. "those who were stateless" (line 61)
  4. Some sentences are long and could be rewritten to clarify the meaning, i.e. line 75-78.

Author Response

Stockholm, October 24th 2020

Dear Editor,

We are extremely grateful for the positive feedbacks and comments from the reviewers!

The suggested comments gave us the opportunity in these days to improve the quality of the manuscripts.

We addressed all points raised by reviewers, please find below a detailed point -by- point   response to reviewers comments.

In particular, with respect to comments by Reviewer 1:

1. The authors should include the scientific named gives to Covid-19: SARS-CoV-2.

Response. Thank you for this comment. We included the scientific name SARS- CoV- 2 in order to clarify difference between the virus that causes the disease and disease itself i.e. Covid-19 (In Introduction, lines 1-2)

2. The key points or definition of a positive multicultural immigrant integration should be included.

Response: We agree that the definition of a positive multicultural immigrant integration make the whole context more comprehensive. Therefore, we added a definition of a positive multicultural immigrant integration, as it is postulated  by Kirmayer, 2019, as the model which consider : “ cultural diversity a public good and a potential source of richness for civil society”. We also believe that this definition can clarify the context.

2. Examples should be included in some sentence, i.e. "those who were stateless" (line 61)

Response: We included example as suggested. We believe that it is worthy to mention that majority of those who are, or in this moment were, stateless are coming from Palestine (line 65).

3. Some sentences are long and could be rewritten to clarify the meaning, i.e. line 75-78.

Response: We re-conceptualised the named sentences in order to make them more clear and comprehensive to readers. We tried to omit some unnecessary parts and rewrite all other sentences in the manuscript for better reader clarity. The manuscript has been proof-read by a native speaker.

Sincerely,

Giuseppe Valeriani

Reviewer 2 Report

General comments

This is a timely commentary on disparities in infection rate and mortality in ethnic minorities during the COVID-19 pandemic. There are some interesting suggestions presented regarding the role digital health solution could play, as well as potential barriers among ethnic minorities.

Specific comments

Introduction

  • No changes required.

Socio-demographic data

  • The incidence of COVID-19 by country of birth are reported only until 7 May 2020. Is there a reason why the data are not reported after this time point? Where there any changes in the incidence rate later in the progression of the pandemic?
  • The results are compared between different countries of birth without any statistical analysis. A 95% confidence interval should be calculated using the formula of Gardner and Altman and significant differences reported:
    • Gardner, M. J. and Altman, D. G. (1986) 'Confidence-intervals rather than P values: estimation rather hypothesis testing', British Medical Journal, 292(6522), pp. 746-750.
  • The same analysis should be carried out for the mortality rates.

Digital health

  • Do the authors have any information on the effectiveness of the initiatives mentioned as taking place since the end of April 2020? It would be interesting to know whether the greater incidence of COVID-19 reported in Table 1 changed after the introduction of the culturally and linguistically appropriate platforms.

Author Response

Stockholm, October 24th 2020

Dear Editor,

We are extremely grateful for the positive feedbacks from the reviewers!

The suggested comments gave us the opportunity in these days to improve the quality of the manuscripts.

We addressed all points raised by reviewers, please find below a detailed point -by- point   response to reviewers comments.

In particular, with respect to comments by Reviewer 2:

1. The incidence of COVID-19 by country of birth are reported only until 7 May 2020. Is there a reason why the data are not reported after this time point? Where there any changes in the incidence rate later in the progression of the pandemic?

Response: We are thankful to reviewer for making this point as it allows us to clarify that we depended on the work of Swedish Public Health Agency (SPHA) regarding collecting the data. We were gently provided with the data related to the difficult first wave of Covid-19, March- May 2020 and all data are public and can be reached on SPHAs official website. SPHA is currently working on a new report, which will be published on SPHA´s official site in the future, as we have been informed in their answer on our e-mails request on 22 October. As we also consider this as a sort of limitation, we explained this in section Considerations and future directions (lines 219-222).

2. The results are compared between different countries of birth without any statistical analysis. A 95% confidence interval should be calculated using the formula of Gardner and Altman and significant differences reported: Gardner, M. J. and Altman, D. G. (1986) 'Confidence-intervals rather than P values: estimation rather hypothesis testing', British Medical Journal, 292(6522), pp. 746-750. The same analysis should be carried out for the mortality rates.

Response: Based on reviewer´s suggestion we added the calculated confidence interval (C.I) as they had been reported in cited articles (between line 116 and 132).

3. Do the authors have any information on the effectiveness of the initiatives mentioned as taking place since the end of April 2020? It would be interesting to know whether the greater incidence of COVID-19 reported in Table 1 changed after the introduction of the culturally and linguistically appropriate platforms.

Response: We are tankful to the reviewer for making this point of view. This manuscript shows our perspective on current situation and we tried to point out importance of cultural appropriate measures as we truly believe that there is no public health without considering those who are in unfavourable situation, such as migrants and/or other minority groups. In this moment, to the best of our knowledge, there are no available date on effectiveness of mentioned initiatives because no research has been conducted on them so far. We believe this data are going to be available in future together with the data related to the period after May 2020. We have added this comments in the section Considerations and further directions (lines 272-280). Please also see our response above under # 1.

Sincerely,

Dr. Giuseppe Valeriani
